# SORSA: Singular Values and Orthonormal Regularized Singular Vectors Adaptation of Large Language Models

## Abstract

In this paper, we propose Singular Values and Orthonormal Regularized Singular Vectors Adaptation, or SORSA, a novel PEFT method. Each SORSA adapter consists of two main parts: trainable principal singular weights $W_p = U_p\text{diag}(S_p)V_p^\top$, and frozen residual weights $W_r = U_r\text{diag}(S_r)V_r^\top$. These parts are initialized by performing singular value decomposition (SVD) on pre-trained weights. Moreover, we implement and analyze an orthonormal regularizer, which we prove could decrease the condition number of $W_p$ and make the optimization more efficient. SORSA adapters could be merged during inference, thus eliminating any inference latency. We also introduce a method to analyze the variation of the parameters by performing SVD and discuss and analyze SORSA's superiority in minimizing the alteration in the SVD aspect. After all, SORSA shows a faster convergence than LoRA and PiSSA in our experiments. On the GSM-8K benchmark, Llama 2 7B adapted using SORSA achieved 56.03% accuracy, surpassing LoRA (42.30%), AdaLoRA (47.30%), Full FT (49.05%), and PiSSA (53.07%). On the MATH benchmark, SORSA achieved 10.36% accuracy, outperforming LoRA (5.50%), AdaLoRA (6.48%), Full FT (7.22%), and PiSSA (7.44%). We conclude that SORSA offers a new perspective on parameter-efficient fine-tuning, demonstrating remarkable performance.

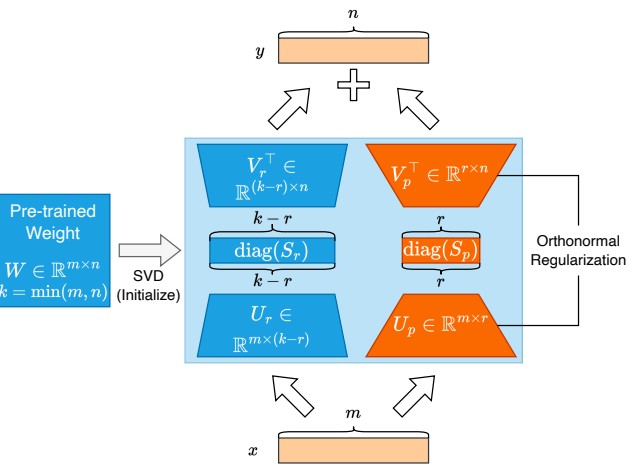

Figure 1: **Architecture of a SORSA adapter.** We only train parts rendered in orange ($U_p$, $\text{diag}(S_p)$ and $V_p^\top$), and freeze parts rendered in blue ($U_r$, $\text{diag}(S_r)$ and $V_r^\top$).

# 1 INTRODUCTION

Pre-trained large language models (LLMs) show remarkable generalization abilities, allowing them to perform various kinds of natural language processing (NLP) tasks (Peng et al., 2024; Touvron et al., 2023; Dubey et al., 2024; Radford et al., 2019; OpenAI, 2023). For specific downstream tasks, full parameter fine-tuning, which continues training all parameters of LLMs on downstream data, is widely used.

However, as the number of parameters in LLMs rapidly increases, full parameter fine-tuning becomes increasingly inefficient. For example, the estimated VRAM requirement for fully fine-tuning Llama 2 7B using Float32 could approach approximately 100 GB, making it unlikely to fully fine-tune the model on a single GPU with current technology. Additionally, the VRAM requirement for fully fine-tuning Llama 2 70B using Float32 exceeds 1 TB (Touvron et al., 2023; Anthony et al., 2023), thus rendering it unfeasible on a single GPU with current technology.

To address these challenges, several parameter-efficient fine-tuning (PEFT) methods (Houlsby et al., 2019; Lester et al., 2021; Hu et al., 2021) have been proposed. These methods enable the training of only a few parameters, significantly reducing VRAM requirements while achieving comparable or even superior performance to full fine-tuning. For instance, tuning Llama 2 7B in Float32 by LoRA (Hu et al., 2021) with a rank of 128 only takes approximately 60GB VRAM, which allows training on $1 \times$ NVIDIA A100 (80GB), or even $3 \times$ NVIDIA RTX 4090 (24GB).

Among those PEFT methods, LoRA (Hu et al., 2021) and its variants (Zhang et al., 2023; Meng et al., 2024; Liu et al., 2024; Dettmers et al., 2024) had become increasingly popular due to their: 1. Low training VRAM requirement 2. No inference latency 3. Versatility in different neuron network architectures.

This paper proposes a novel PEFT approach, Singular Values and Orthonormal Regularized Singular Vectors Adaptation, or **SORSA**. A SORSA adapter has two main parts: principal singular weights $W_p = U_p \text{diag}(S_p) V_p^\top$, and residual weights $W_r = U_r \text{diag}(S_r) V_r^\top$. These two parts are initialized by performing singular value decomposition (SVD) on pre-trained weight. Residual singular values and vectors will be merged into one matrix and frozen while training. We only train principal singular values and vectors with an orthonormal regularizer implemented to keep the orthonormality of $U_p$ and $V_p^\top$. The architecture of a SORSA adapter is illustrated in Figure 1.

Furthermore, we analyze the pattern of variation of singular values and vectors during parameter updating and discuss the different patterns of fine-tuning (FT), LoRA, SORSA without regularizer, and SORSA with regularizer concerning singular values and vectors' updating.

We also provide a comprehensive gradient analysis with a mathematical foundation for SORSA. This analysis demonstrates several crucial properties of our method, including the convexity of the regularizer, Lipschitz continuity of the gradient, and bounds on the hyperparameter $\gamma$. Moreover, we prove that SORSA improves the condition number of the optimization problem compared to unregularized approaches.

SORSA retains all the benefits of LoRA and its variants while demonstrating remarkable performance compared to PiSSA, LoRA, and full parameter fine-tuning in our experiments.

# 2 RELATED WORKS

Parameter-efficient fine-tuning (PEFT) methods have been developed to address the inefficiency of full parameter fine-tuning for large language models. These methods focus on adapting the model for downstream tasks while updating only a few parameters and keeping most of the model's weights frozen. This approach significantly reduces the memory and computational requirements during training, especially VRAM.

## 2.1 ADAPTER-BASED PEFT

Adapter-based PEFT methods are the first type of PEFT initially designed by Houlsby et al. (2019). It introduces additional trainable non-linear blocks into the frozen pre-trained model, which could effectively tune the pre-trained model with a limited amount of trainable parameters. Its variants,

e.g., Lin et al. (2020), reduce the number of adapter layers per block, and He et al. (2022) focus on adding adapter modules parallel to existing layers. However, all adapter-based PEFT methods introduce inference latency due to their non-mergeable attribute.

## 2.2 PROMPT-BASED PEFT

Prompt-based PEFT is a well-known PEFT type first proposed in Lester et al. (2021). This work has several variants, including Liu et al. (2022a); Razdaibiedina et al. (2023). However, they have some inevitable shortcomings, such as potential performance limitations compared to full parameter fine-tuned models, additional inference latency due to expanding the length of the total input to the model, and the complexity of designing effective initialization.

## 2.3 LoRA AND ITS VARIANTS

LoRA (Hu et al., 2021) and its variants are the most popular type of PEFT methods. This type of PEFT is popular due to its on-par or better performance than full parameter fine-tuning without introducing any inference latency. LoRA could be represented by equation $W = W_0 + BA$, where $W_0 \in \mathbb{R}^{m \times n}$ is the pre-trained weight, $A \in \mathbb{R}^{m \times r}$, using Gaussian initialization, and $B \in \mathbb{R}^{r \times n}$, using zero initialization, are low-rank matrices.

Its variant, for example, AdaLoRA (Zhang et al., 2023), introduces an SVD decomposition and pruning for least significant singular values for more efficient parameter updating.

DoRA (Liu et al., 2024) proposed a novel way to decompose weight into direction and magnitude by $W = \underline{m} \frac{W_0 + \underline{BA}}{\|W_0 + \underline{BA}\|_c}$, where $\underline{m}$ is initialized by $\underline{m} = \|W_0 + \underline{BA}\|_c$, $\| \cdot \|_c$ denotes column-wise norm. The results show that DoRA has a better learning capacity than LoRA. However, DoRA introduced a calculation of norms in every training step, which makes it much more inefficient than LoRA.

OLoRA (Büyükakyüz, 2024) uses QR decomposition to initialize the LoRA adapters $A$ and $B$, which initializes $B$ as an orthogonal matrix. They discuss the significance of orthonormality in neural networks' weight (See Section 5 for more details). In their experiments, OLoRA demonstrates faster convergence than LoRA.

PiSSA (Meng et al., 2024) decomposes pre-trained weight $W_0 = U\text{diag}(S)V^\top$ by Singular Value Decomposition (SVD) and then splits $W_0$ into $W_{pri}$ and $W_{res}$: $W_{pri} = AB$ which is trainable. Using PyTorch (Paszke et al., 2019) split notation, $A$ and $B$ are defined by $A = U_{[:,:r]}\text{diag}(S_{[:r]}^{\frac{1}{2}})$ and $B = \text{diag}(S_{[:r]}^{\frac{1}{2}})V_{[:,r,:]}^\top$; $W_{res} = U_{[:,r:]}\text{diag}(S_{[r:]})V_{[r:,:]}^\top$ which is frozen. PiSSA results in a faster convergence speed and better fitting than LoRA.

SORSA's architecture is similar to PiSSA, which conducts SVD and replaces pre-trained weights with residual singular weights. SORSA also adopted the regularizer present in AdaLoRA. In general, SORSA inherits LoRA and its variants' benefits, including low training VRAM requirement, no inference burden, and versatility in different architectures.

## 2.4 OTHER METHODS

There are also a few efficient adapting methods with unique techniques. For example, GaLore (Zhao et al., 2024) is a memory-efficient PEFT method that reduces VRAM usage by leveraging gradient accumulation and low-rank approximation. LISA (Pan et al., 2024) uses a layer-wise importance sampling approach, prioritizing layers significantly impacting model performance and selectively fine-tuning essential parameters.

# 3 SORSA: SINGULAR VALUE AND ORTHONORMAL REGULARIZED SINGULAR VECTOR ADAPTATION

Giving a matrix $W \in \mathbb{R}^{m \times n}$, let $k = \min(m, n)$, we could perform SVD to decompose $W$ by $W = U\text{diag}(S)V^\top$. Here, $U \in \mathbb{R}^{m \times k}$ is a matrix of left singular vectors and has orthonormal columns, $V \in \mathbb{R}^{n \times k}$ is a matrix of right singular vectors and has orthonormal columns, and $S \in \mathbb{R}^k$

are singular values $\sigma^1, \sigma^2 \ldots \sigma^k$ arranged in descending order. diag(S) is constructed by placing the elements of $S \in \mathbb{R}^k$ along the main diagonal, with all other elements zero.

According to our SVD notations, given a rank $r$ where $r \ll k$, we could perform the low-rank approximation by selecting the first $r$ items on the diagonal of $\Sigma$, which is the first $r$ most significant singular values, and also select the first $r$ columns of $U$ and first $r$ rows of $V^\top$, which correspond to the selected singular values. By performing SVD low-rank approximation, we could get a low-rank matrix that preserves the largest significant values and vectors, containing the matrix's "most essential" data.

Therefore, for a pre-trained weight $W_0 \in \mathbb{R}^{m \times n}$, we could split it based on its singular value into principal weight $W_p$ and residual weight $W_r$, where $W_p$ contains the most important part of information of the matrix, and $W_r$ contains the least significant part

$$W_p = U_{[:,:r]}\text{diag}(S_{[:r]})V_{[:r,:]}^\top \in \mathbb{R}^{m \times n}; \tag{1}$$

$$W_r = U_{[:,r:]}\text{diag}(S_{[r:]})V_{[r:,:]}^\top \in \mathbb{R}^{m \times n}. \tag{2}$$

Here, $U$ represents the matrix of left singular vectors, $S$ represents the singular values, diag$(W)$ denotes a function to form a diagonal matrix from $W$, and $V$ represents the matrix of right singular vectors. We use PyTorch (Paszke et al., 2019) syntax to demonstrate matrix selection, where $[:, : r]$ denotes selecting the first $r$ columns of the matrix, and $[r :, :]$ denotes selecting the last $r$ rows of the matrix. We rewrite $U_{[:,:r]}$, $S_{[:r]}$ and $V_{[:r,:]}^\top$, which constitute $W_p$, as $U_p$, $S_p$ and $V_p^\top$ for simplicity, and rewrite $U_{[:,r:]}$, $S_{[:r]}$ and $V_{[r:,:]}^\top$, which constitute $W_r$, as $U_r$, $S_r$ and $V_r^\top$ correspondingly.

The initialization of $W_r$ in SORSA is the same as PiSSA (Meng et al., 2024). Nevertheless, unlike PiSSA which merge diag$(S_p)$ with $U_p$ and $V_p^\top$ into $A$ and $B$ by $A = U_p\text{diag}(S_p)^{\frac{1}{2}}$ and $B = \text{diag}(S_p)^{\frac{1}{2}}V_p^\top$, SORSA remains $U_p$, $S_p$, and $V_p^\top$ in separate matrices. SORSA is defined by Equation (3), initially equivalent to the pre-trained weight $W_0$.

During training, $W_r$ remains frozen, and only $U_p$, $S_p$, and $V_p^\top$ are updated.

SORSA is defined as

$$\text{SORSA}(x) := x(W_r + W_p) = xW_r + xU_p\text{diag}(S_p)V_p^\top. \tag{3}$$

In our implementation, we use an optimized version of the SORSA equation, which results in a much faster computation speed. See Appendix A for more details.

We adopt an orthonormal regularizer similar to (Zhang et al., 2023) for $U_p$ and $V_p^\top$

$$\mathcal{L}_{reg} = \|U_p^\top U_p - I\|_F + \|V_p^\top V_p - I\|_F, \tag{4}$$

where $\mathcal{L}_{reg}$ is the orthonormal regularizer loss, the $U_p$ and $V_p^\top$ are each orthonormal vectors in columns and rows, respectively, after initialization due to SVD's property. The regularizer could enhance their orthonormality during training. We discuss and verify its importance and effectiveness in Sections 4 and 5.

Therefore, parameter updating of $W_p$ in a SORSA adapter at training step $t$ could be expressed as:

$$W_{p,t+1} = W_{p,t} - \eta_t \nabla_{W_{p,t}}\mathcal{L}_{train} - \gamma_t \nabla_{W_{p,t}}\mathcal{L}_{reg}. \tag{5}$$

At training step $t$, $\nabla_{W_{p,t}}\mathcal{L}_{train}$ denotes the gradient of $\mathcal{L}_{train}$ respect to $W_{p,t}$, and $\nabla_{W_{p,t}}\mathcal{L}_{reg}$ denotes the gradient of the orthonormal regularizer loss $\mathcal{L}_{reg}$ respect to $W_{p,t}$. $\eta_t$ and $\gamma_t$ are the learning rates for training loss and regularizer loss at step $t$, respectively.

We update the SORSA as the following for implementation simplicity

$$W_{p,t+1} = W_{p,t} - \eta_t \left( \nabla_{W_{p,t}}\mathcal{L}_{train} + \frac{\gamma}{\eta_d} \nabla_{W_{p,t}}\mathcal{L}_{reg} \right), \tag{6}$$

$\eta_d$ is the maximum learning rate from the scheduler. This implementation allows us to use only one optimizer and scheduler to deal with two different learning rates separately.

# 4 SINGULAR VALUES AND VECTOR ANALYSIS

## 4.1 ANALYSIS METHOD

The study of DoRA (Liu et al., 2024) introduces an analysis method that focuses on the deviation of magnitude and direction ($\Delta M$, $\Delta D$) during training of full parameter fine-tuning and LoRA (Hu et al., 2021). They discovered that the distinction between full parameter fine-tuning and LoRA likely affects their learning ability difference. Inspired by their methods, we propose a novel technique that analyzes the correlation between the deviation of singular values ($\Delta \Sigma$) and singular vectors ($\Delta D$) from pre-trained matrices during updating. Our analysis suggests a significant difference in singular values and vectors' stability and an updating pattern of fine-tuning, LoRA, and SORSA.

The singular value and vector variations between pre-trained weight $W_0 \in \mathbb{R}^{m \times n}$ and tuned weight $W_t \in \mathbb{R}^{m \times n}$, which $t$ denotes the training step, could be defined as follows

$$\Delta \Sigma_t = \frac{\Sigma_{i=1}^k \left| \sigma_t^i - \sigma_0^i \right|}{k}, \tag{7}$$

where $\Delta \Sigma_t$ represents the singular value difference between $W_0$ and $W_t$ at training step $t$. $\sigma_t^i$ denotes the $i$-th element in diagonal of $\Sigma_t$, where $\Sigma_t$ is decomposed from $W_t$ by performing SVD, $k = \min(m, n)$,

$$\Delta U_{t,j} = \left| \langle \mathbf{u}_t^j, \mathbf{u}_0^j \rangle \right|; \tag{8}$$

$$\Delta V_{t,i}^\top = \left| \langle \mathbf{v}_t^i, \mathbf{v}_0^i \rangle \right|; \tag{9}$$

$$\Delta D_t = 1 - \frac{1}{2k} \sum_{i=0}^k (\Delta U_{t,i} + \Delta V_{t,i}^\top). \tag{10}$$

Here, $k = \min(m, n)$; $\mathbf{u}_t^j$ denotes the $j$-th column vector of matrix $U_t$, and $\mathbf{v}_t^i$ denotes the $i$-th row vector of matrix $V_t^\top$; $\Delta D_t \in (0, 1)$ represents variation of singular vectors between $W_0$ and $W_t$ at training step $t$; $U_t$ and $V_t^\top$ are decomposed from $W_t$ by performing SVD.

We adopt the analysis on Llama 2 7B (Touvron et al., 2023) using the first 100K data of Meta-MathQA (Yu et al., 2024). We test fine-tuning, LoRA, and SORSA (with and without regularizer). See Appendix B.1 for training details of the analysis.

## 4.2 ANALYSIS RESULT

Based on our collected data, this section analyzes the results of different training methods: fine-tuning, LoRA, and SORSA. The analysis data is illustrated in Figure 2.

Based on our collected data, we analyze how different training methods - partial fine-tuning, LoRA, and SORSA (with and without regularizer) - affect the pre-trained weights' structure and information preservation.

The analysis reveals several key insights about how these methods interact with the pre-trained knowledge:

1. Partial fine-tuning and LoRA show substantial alterations in singular vectors (large $\Delta D$), indicating significant disruption to the fundamental structure of the pre-trained weights. This extensive modification likely damages the model's carefully learned generalizations across multiple domains, leading to catastrophic forgetting. The parallel updating patterns across different layers suggest these methods make broad, potentially destructive changes throughout the model rather than targeted adaptations.

2. SORSA with regularizer demonstrates significantly smaller changes in both singular values ($\Delta \Sigma$) and singular vectors ($\Delta D$) compared to other methods. This controlled modification suggests that SORSA better preserves the pre-trained model's underlying knowledge structure while making precise adjustments for the downstream task. The orthonormal regularizer appears to act as a constraint that helps maintain the original geometric relationships within the weight matrices that encode the model's generalized capabilities.

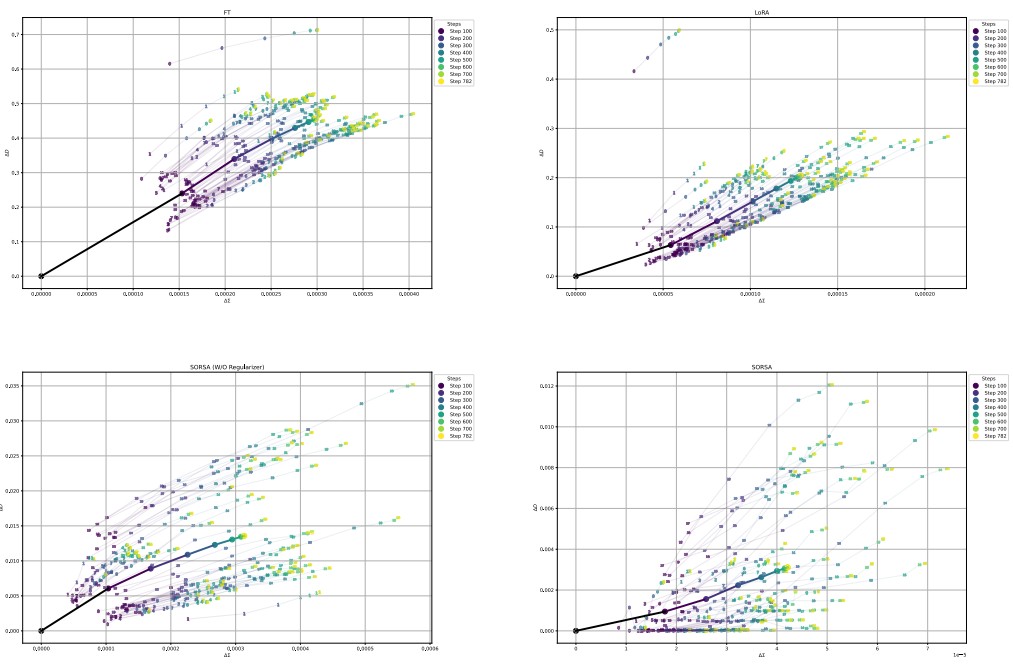

Figure 2: $\Delta D$ **and** $\Delta\Sigma$ **of each trainable parameters during training steps.** Numbers in the plot represent layer of the weight. Dots represent mean $\Delta D$ and $\Delta\Sigma$ at specific step.

3. Different matrices in SORSA show distinct, non-parallel updating patterns, unlike the uniform changes seen in other methods. This suggests SORSA can identify and selectively modify the most relevant components for the target task while leaving other capabilities largely intact. This targeted adaptation explains why SORSA can achieve better performance with less disruption to the model's general knowledge.

4. When SORSA is used without the orthonormal regularizer, we observe larger changes in both $\Delta D$ and $\Delta\Sigma$, along with more uniform updating patterns similar to LoRA and partial fine-tuning. This empirically validates the regularizer's crucial role in preserving the pre-trained model's information structure while allowing efficient adaptation.

These patterns indicate SORSA's ability to preserve the rich, generalized knowledge embedded in pre-trained weights while making precise adjustments for specific tasks. This property enables higher learning rates without over-fitting and explains SORSA's improved performance across various benchmarks. The method's ability to maintain the model's fundamental structure while allowing targeted modifications represents a significant advance in efficient model adaptation.

## 5 GRADIENT ANALYSIS

In this section, we present a comprehensive mathematical analysis of the SORSA method, which mainly focuses on the effect of orthonormal regularization. Our investigation elucidates the fundamental optimization properties of SORSA, providing a theoretical foundation for its advantages. We explore four critical aspects: the convexity of the regularizer, the Lipschitz continuity of the gradient, bounds on the hyperparameter $\gamma$, and the impact on the condition number of the optimization problem.

The proofs of the theorems and lemmas and additional mathematical details are provided in Appendix C.

Our analysis reveals the fundamental theoretical properties of SORSA, establishing its mathematical soundness and demonstrating its optimization advantages. We prove two key theorems that form the cornerstone of our theoretical framework.

**Theorem 5.1.** *The regularizer $\mathcal{L}_{reg}$ is convex.*

**Theorem 5.2.** *The gradient of the regularizer $\mathcal{L}_{reg}$ is Lipschitz continuous.*

Building upon these foundational results, we further analyze the bounds of the hyperparameter $\gamma$, a critical factor in the performance of SORSA:

**Theorem 5.3.** *For convergence of gradient descent, the learning rate $\eta_d$ and regularization parameter $\gamma$ should*

$$\gamma \propto \frac{1}{\eta_d}. \tag{11}$$

This theorem provides crucial guidance for practitioners, offering an explicit criterion for selecting appropriate values of $\gamma$ to ensure convergence of the gradient descent process.

To demonstrate SORSA's superior optimization properties, we present a novel analysis of condition numbers during the optimization process, a critical factor in determining convergence speed and stability. Our theoretical investigation reveals a significant improvement in the condition number compared to unregularized approaches, providing a mathematical foundation for SORSA's enhanced performance.

We begin this analysis by establishing a key lemma that bounds the effect of the orthonormal regularizer on the singular values of the weight matrix:

**Lemma 5.4.** *Let $W_p^{unreg} = U_p^{unreg} diag(S_p)^{unreg} (V_p^{unreg})^\top$ be the $W_p$ only training without using regularizer, and $W_p^{reg} = U_p^{reg} diag(S_p)^{reg} (V_p^{reg})^\top$ be the $W_p$ training with the regularizer. For each singular value $\sigma_i$, the following bound holds:*

$$(1 - \epsilon)\sigma_i^{unreg} \leq \sigma_i^{reg} \leq (1 + \epsilon)\sigma_i^{unreg}, \tag{12}$$

*where $\epsilon$ is a small positive constant, $\sigma_i^{reg}$ and $\sigma_i^{unreg}$ are singular values in the case of training with and without regularizer, respectively.*

Lemma 5.4 provides a crucial connection between the regularizer and the singular values. Building on this result, we arrive at our main theorem regarding the condition number:

**Theorem 5.5.** *The orthonormal regularizer in SORSA can improve the condition number of the optimization problem throughout training under certain conditions. Specifically, at initialization ($t = 0$):*

$$\kappa(W_{p,0}^{reg}) = \kappa(W_{p,0}^{unreg}), \tag{13}$$

*where $\kappa(W_p)$ denotes the condition number of $W_p$; $W_{p,t}^{reg}$ and $W_{p,t}^{unreg}$ represent $W_p$ at time-step $t$ in the case of training with or without regularizer, respectively.*

*$\exists c > 0$, while $t > c$,*

$$\kappa(W_{p,t}^{reg}) < \kappa(W_{p,t}^{unreg}). \tag{14}$$

This theorem quantifies the improvement in the condition number achieved by SORSA, offering an explanation for its fast convergence. The proof leverages the effects of the orthonormal regularization to establish a tight bound on the condition number ratio. This theorem could also show that training with the regularizer, the distribution of $W_p$ will be more evenly distributed due to a smaller ratio between $\sigma_{\max}(W_p)$ and $\sigma_{\min}(W_p)$, which means better training stability.

Moreover, as mentioned in Büyükakyüz (2024), orthonormal matrices in neuron networks could improve gradient flow (Saxe et al., 2014; Arjovsky et al., 2016) and enhanced optimization landscape (Huang et al., 2018; Wisdom et al., 2016), which could also explain SORSA's superior performance in convergence.

In conclusion, these theorems provide a mathematical foundation for the SORSA method. These theoretical guarantees validate SORSA's empirical success and provide valuable insights for future developments in PEFT methods.

## 6 EMPIRICAL EXPERIMENTS

We conducted comparative experiments on different NLP tasks, including natural language generation (NLG) between SORSA, PiSSA (Meng et al., 2024), LoRA (Hu et al., 2021), and full parameter fine-tuning.

We conducted NLG tests on Llama 2 7B (Touvron et al., 2023), RWKV6 7B (Peng et al., 2024), Mistral 7B v0.1 (Jiang et al., 2023) Gemma 7B (Gemma Team et al., 2024). We trained the models using the first 100K data in MetaMathQA (Yu et al., 2024) and evaluated the model on GSM-8K (Cobbe et al., 2021) and MATH (Hendrycks et al., 2021). We also trained the model on the first 100K data in CodeFeedback Filtered Instruction (Zheng et al., 2024) dataset and evaluated it on HumanEval (Chen et al., 2021). The training process followed identical setups as the experiments conducted in PiSSA (Meng et al., 2024). All reported values are accuracy in percentage. See Appendix B.2 for more details and hyperparameters of the training. We quoted some PiSSA, LoRA, and full parameter fine-tuning results from Meng et al. (2024). Some of our experiments were conducted on a single NVIDIA A100-SXM4 (80GB) GPU, and others were conducted on a single NVIDIA H100-SXM4 (80GB) GPU. See Table 1 for the results and Figure 3 for the loss and gradient norm comparison.

| Model | Method | Trainable Parameters | GSM-8K | MATH | HumanEval |
|---|---|---|---|---|---|
| Llama 2 7B | Full FT | 6738M | 49.05[†] | 7.22[†] | 21.34[†] |
|  | LoRA | 320M | 42.30[†] | 5.50[†] | 18.29[†] |
|  | PiSSA | 320M | 53.07[†] | 7.44[†] | 21.95[†] |
|  | AdaLoRA | 320M | 47.30 | 6.48 | 19.51 |
|  | SORSA | 320M | **56.03** | **10.36** | **24.39** |
| RWKV6 7B | LoRA | 176M | 8.04[1] | 7.38 | 15.24 |
|  | PiSSA | 176M | 32.07 | 9.42 | 17.07 |
|  | AdaLoRA | 176M | 33.28 | 8.08 | 15.85 |
|  | SORSA | 176M | **45.87** | **11.32** | **22.56** |
| Mistral 7B | Full FT | 7242M | 67.02[†] | 18.60[†] | 45.12[†] |
|  | LoRA | 168M | 67.70[†] | 19.68[†] | 43.90[†] |
|  | PiSSA | 168M | 72.86[†] | 21.54[†] | 46.95[†] |
|  | AdaLoRA | 168M | 72.25 | 21.06 | 45.73 |
|  | SORSA | 168M | **73.09** | **21.86** | **47.56** |
| Gemma 7B | Full FT | 8538M | 71.34[†] | 22.74[†] | 46.95[†] |
|  | LoRA | 200M | 74.90[†] | 31.28[†] | 53.66[†] |
|  | PiSSA | 200M | 77.94[†] | **31.94** [†] | 54.27[†] |
|  | AdaLoRA | 200M | **78.99** | 31.44 | **55.49** |
|  | SORSA | 200M | 78.09 | 29.52 | **55.49** |

Table 1: Comparing SORSA with other methods on NLG tasks. [†] denotes results from Meng et al. (2024).

The results showed that across all models tested, SORSA generally outperformed other methods, though with some notable exceptions. For mathematical evaluations on Llama 2 7B, SORSA scored 56.03% on GSM-8K and 10.36% on MATH, significantly outperforming other methods. For the RWKV6 7B model, SORSA achieved 45.87% accuracy on GSM-8K and 11.32% on MATH, surpassing both PiSSA and AdaLoRA, with AdaLoRA showing competitive performance on GSM-8K at 33.28%. On Mistral 7B, SORSA reached 73.09% on GSM-8K and 21.86% on MATH, showing modest improvements over AdaLoRA's strong performance of 72.25% and 21.06%, respectively. With Gemma 7B, the results were mixed - while AdaLoRA achieved the highest GSM-8K score at 78.99% and competitive MATH performance at 31.44%, SORSA maintained strong performance with 78.09% on GSM-8K. However, its MATH score of 29.52% was lower than other methods. In

---

[1]This significant under-perform due to LoRA failed to learn the GSM-8K required answer formatting behavior.

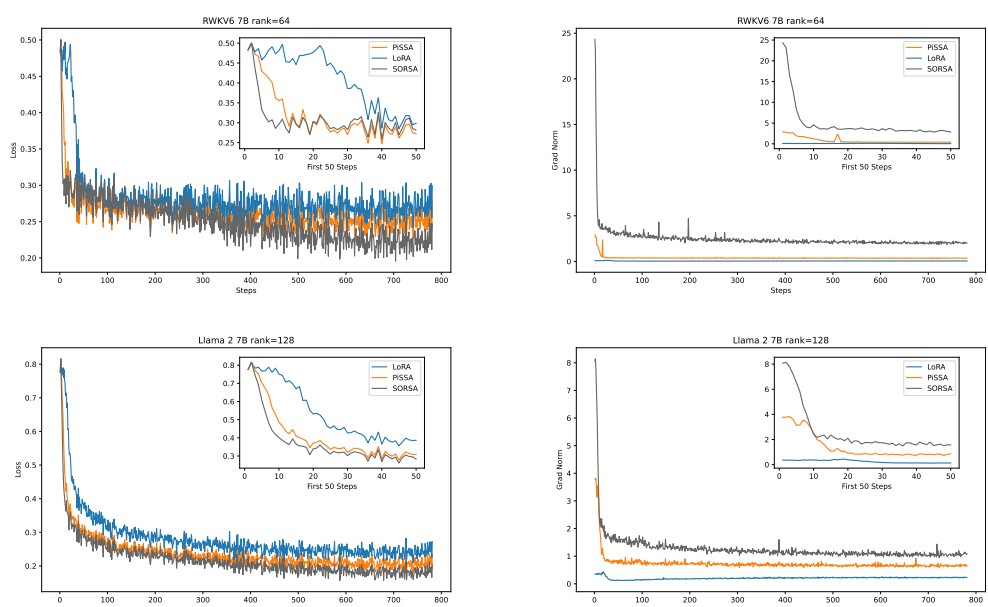

Figure 3: The training loss and gradient norm comparison between SORSA, PiSSA, and LoRA on MetaMathQA training of RWKV6 7B and Llama 2 7B. LoRA and PiSSA curves of Llama 2 7B are from Meng et al. (2024).

coding evaluations, SORSA and AdaLoRA showed strong performance on HumanEval, with both methods achieving 55.49% on Gemma 7B, while SORSA maintained an edge across other model variants. Additionally, we did not include loss and gradient norm curves in our figure because the regularizer in AdaLoRA and Gaussian initialization caused significantly higher initial loss values, making direct comparisons with other methods inappropriate.

The Figure 3 reveals that SORSA and PiSSA exhibit nearly identical loss curves at the beginning and even slightly higher than PiSSA on RWKV-6 training. However, when the training step is approximately $t > 300$, SORSA steadily decreases its loss. In contrast, LoRA and PiSSA show a deceleration in their loss reduction. The observations on loss curves are also valid for the changing rate of gradient norm, where SORSA showed a more consistent decrease in gradient norm compared to LoRA and PiSSA. This supports Theorem 5.5, especially at later stages of training.

However, due to the limitation of computing resources, we only trained and benchmarked a small number of tasks.

## 7 CONCLUSION

In this paper, we introduced SORSA, a novel parameter-efficient fine-tuning (PEFT) method designed to enhance the adaptation of large language models (LLMs) for downstream tasks. SORSA utilizes singular value decomposition (SVD) to split pre-trained weights into principal and residual components, only training the principal singular values and vectors while freezing the residuals. We implemented an orthonormal regularizer to maintain the orthonormality of singular vectors during training, ensuring efficient parameter updates and preserving the integrity of singular values.

Our experiments demonstrated that SORSA outperforms existing PEFT methods, such as LoRA and PiSSA, in both convergence speed and accuracy on the NLG tasks. Specifically, Llama 2 7B, tuned with SORSA, achieved significant improvements in the GSM-8K and MATH benchmarks, highlighting the effectiveness of our approach.

We adopted singular values and vector analysis, comparing SORSA with FT and LoRA. SORSA is superior in preserving the pre-trained weight's singular values and vectors during training. This

suggests an explanation for SORSA's supreme performance demonstrated in the experiment. We also show the significance of the orthonormal regularizer through analysis.

Our gradient analysis provided a mathematical foundation for SORSA, demonstrating its convexity, Lipschitz continuity, and the crucial role of the regularizer in improving the optimization landscape. This theoretical framework explains SORSA's empirical superior performance and offers valuable insights for future developments in adaptive learning algorithms.

SORSA retains the advantages of LoRA and variants, including low training VRAM requirements, no inference latency, and versatility across different neural network architectures. By offering a more efficient fine-tuning mechanism, SORSA presents a promising direction for future research and application in the field of LLMs.

Overall, SORSA gives a new perspective on parameter-efficient fine-tuning, showcasing exceptional efficiency and robust performance. It outperforms existing methods like LoRA and PiSSA in several downstream tasks and maintains the practical benefits of low VRAM requirements, no inference latency, and ease of implementation. This innovative approach offers a promising direction of singular values and vector analysis for future research and practical applications in adapting pre-trained models, making it a pivotal development in the field.

## 8 FUTURE WORK

While SORSA demonstrates improvements over existing PEFT methods, several promising directions for future research exist to enhance its capabilities and broaden its impact.

A crucial area for exploration is the application of SORSA beyond natural language processing. While our current evaluation focuses on language models, SORSA's theoretical foundation in singular value decomposition suggests it could be equally effective for computer vision models like Ho et al. (2020); Liu et al. (2022b); Dosovitskiy et al. (2021); Rombach et al. (2022) and multi-modal architectures such as (Radford et al., 2021). Future work should evaluate SORSA's performance on vision transformers, convolutional neural networks, and other architectures across diverse tasks like image classification, object detection, and semantic segmentation. This extended evaluation across different domains would provide valuable insights into SORSA's versatility and potentially uncover domain-specific optimizations.

Another compelling direction is the integration of quantization techniques with SORSA, similar to approaches like QLoRA (Dettmers et al., 2024) and QPiSSA (Meng et al., 2024). Quantization could significantly reduce SORSA's memory footprint and computational requirements while maintaining its efficient adaptation capabilities. This would be particularly valuable for deploying adapted models on edge devices and resource-constrained environments. By combining SORSA's precise parameter updates with the efficiency gains of quantization, we could enable high-quality model adaptation across a much broader range of hardware configurations. This democratization of fine-tuning capabilities could accelerate the adoption of AI technologies in real-world applications, from mobile devices to IoT systems.

By pursuing these research directions, we can build upon SORSA's theoretical foundations to create more versatile and accessible model adaptation techniques. Success in these areas would not only advance the field of parameter-efficient fine-tuning but also help bridge the gap between state-of-the-art AI models and practical applications. This could ultimately lead to more widespread integration of adaptive AI systems across different sectors of society, making advanced machine learning capabilities more accessible and impactful in people's daily lives.

### ETHICS STATEMENT

In this paper, we introduce an innovative PEFT method in machine learning. Our approach significantly streamlined the model's tuning process, particularly for large-scale models, addressing both computational efficiency and environmental sustainability. As we push the boundaries of what is possible with Machine Learning, it is essential to consider the broader impacts of these advancements on the environment and ethical standards within the field.

**Environmental Impact.** Our experiments found that adapting with SORSA could reduce VRAM consumption by up to 80%. This significant reduction in hardware resource requirements also suggests less energy consumption than entire parameter fine-tuning methods. By enhancing efficiency, our approach could significantly reduce the carbon footprint of Machine Learning operations.

**Ethical Concerns.** The PEFT method, while efficient, raises critical ethical concerns regarding the security of built-in safety measures in AI models. As demonstrated in Lermen & Rogers-Smith (2024), subversive fine-tuning techniques can bypass safety training intended to prevent the generation of harmful content. The ease and affordability of such methods underscore the vulnerability of safety protocols. It is imperative to develop robust safeguards that keep pace with technological advancements, ensuring that efficiency gains in model tuning do not compromise the ethical use of AI.

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

# A    FASTER SORSA ADAPTERS

According to the definition of SORSA from Equation (3), because $\text{diag}(S_p)$ is always a diagonal matrix, it is equivalent to:

$$\text{SORSA}(x) = xW_r + x(U_p \odot S_p)V_p^\top, \tag{15}$$

where $\odot$ denotes element-wise multiplication.

This transformation allows us to reduce the computational complexity of SORSA adapters. In the original form, we had to perform matrix multiplication twice. However, in the Equation (15), we only have one matrix multiplication and one element-wise multiplication. The time complexity of $U_p \odot S_p$ is $\mathcal{O}(m \times n)$, much less than complexity of $U_p S_p$, which is $\mathcal{O}(m \times n^2)$. Therefore, while $\lim_{m,n\to\infty}$, the computation speed of SORSA adapters will be the same as LoRA and PiSSA.

We performed a benchmark using PyTorch (Paszke et al., 2019) on an NVIDIA H100 SXM4 (80GB) GPU backed with CUDA and Apple M2 Pro CPU to test the computation time between these two methods. See Figure 4 to see our results.

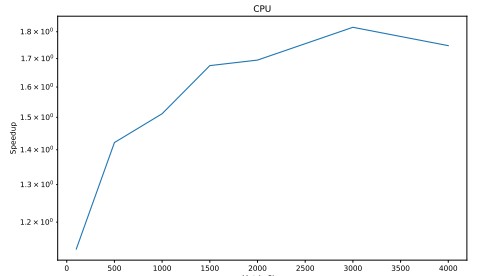 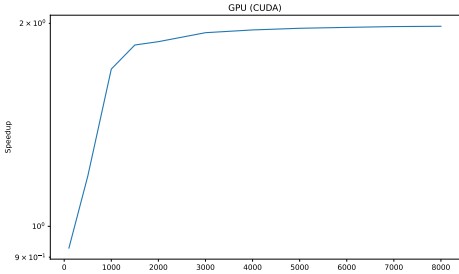

Figure 4: Benchmark between two equations of SORSA

# B EXPERIMENTS DETAILS

## B.1 ANALYSIS

For the singular values and vectors analysis in Section 4, we applied fine-tuning, LoRA and SORSA (with and without orthonormal regularizer) on Llama 2 7B (Touvron et al., 2023) model, training with the first 100K data in MetaMathQA (Yu et al., 2024) dataset. We only calculated the loss on the response part. The models are trained with TF32 & BF16 (Wang & Kanwar, 2019) mix precision. See Table 2 for hyperparameters.

We used AdamW (Loshchilov & Hutter, 2018) optimizer and cosine annealing scheduler in training. In the analysis, LoRA and SORSA were only applied to q_proj and v_proj matrices, respectively. For FT, we set model's q_proj and v_proj matrices to trainable.

We also found we should only perform SVD for analysis using CPU, in order to get the precise analysis data.

| Model | Llama 2 7B | | | |
|---|---|---|---|---|
| Method | FT | LoRA | SORSA (w/o reg) | SORSA |
| Training | | | | |
| Mix-Precision | TF32&BF16 | TF32&BF16 | TF32&BF16 | TF32&BF16 |
| Epoch | 1 | 1 | 1 | 1 |
| Batch Size | 128 | 128 | 128 | 128 |
| Max Length | 512 | 512 | 512 | 512 |
| Weight Decay | 0 | 0 | 0 | 0 |
| Warm-up Ratio | 0.03 | 0.03 | 0.03 | 0.03 |
| Learning Rate | 2e-5 | 2e-5 | 2e-5 | 3e-5 |
| Grad Clip | False | False | False | False |
| SORSA $\gamma$ | N/A | N/A | 0 | 5e-4 |
| Rank | N/A | 128 | 128 | 128 |

Table 2: Hyperparameters for the analysis

## B.2 NLG EXPERIMENTS

For our NLG tasks, we adapted Llama 2 7B (Touvron et al., 2023), RWKV6 7B (Peng et al., 2024), Mistral 7B v0.1 (Jiang et al., 2023) Gemma 7B (Gemma Team et al., 2024) models by SORSA. For GSM-8K (Cobbe et al., 2021) and MATH (Hendrycks et al., 2021) evaluations, we trained those models with the first 100K data in MetaMathQA (Yu et al., 2024) dataset. For HumanEval (Chen et al., 2021) evaluation, we use the first 100K data in CodeFeedback Filtered Instruction(Zheng et al., 2024) dataset.

We used AdamW (Loshchilov & Hutter, 2018) optimizer and cosine annealing scheduler in training. SORSA adapters were applied on all linear matrices in every layer. We only calculated the loss on the response part. The models are loaded in FP32 and trained with TF32 & BF16 mix precision. In our experiments, we selected a higher learning rate for SORSA than other methods to counterbalance the negative effect of orthonormal regularizer on optimizing toward lower training loss. See Table 3 for hyperparameters. See Listing 1 for the prompt we used in GSM-8K and MATH evaluations, and Listing 2 for the prompt we used for HumanEval tests.

| Model | Llama 2 7B | RWKV6 7B | RWKV6 7B | Mistral 7B | Gemma 7B |
|---|---|---|---|---|---|
| Method | SORSA | SORSA | LoRA PiSSA | SORSA | SORSA |
| Training | | | | | |
| Mix-Precision | TF32&BF16 | TF32&BF16 | TF32&BF16 | TF32&BF16 | TF32&BF16 |
| Epoch | 1 | 1 | 1 | 1 | 1 |
| Batch Size | 128 | 128 | 128 | 128 | 128 |
| Max Length | 512 | 512 | 512 | 512 | 512 |
| Weight Decay | 0 | 0 | 0 | 0 | 0 |
| Warm-up Ratio | 0.03 | 0.03 | 0.03 | 0.03 | 0.03 |
| Learning Rate | 3e-5 | 3e-5 | 2e-5 | 3e-5 | 3e-5 |
| Grad Clip | 1.0 | 1.0 | 1.0 | 1.0 | 1.0 |
| SORSA $\gamma$ | 4e-4 | 4e-4 | N/A | 4e-4 | 4e-4 |
| Rank | 128 | 64 | 64 | 64 | 64 |
| Evaluating | | | | | |
| Precision | BF16 | FP32 | FP32 | BF16 | BF16 |
| Sampling | False | | | | |
| Top-P | 1.0 | | | | |
| Max Length | GSM-8K: 1024 MATH: 2048 HumanEval: 2048 | | | | |

Table 3: Hyperparameters of experiments of SORSA, LoRA and PiSSA on different models for GSM-8K and MATH

| Model | Llama 2 7B | Mistral 7B | Gemma 7B | RWKV6 7B |
|---|---|---|---|---|
| Method | AdaLoRA | AdaLoRA | AdaLoRA | AdaLoRA |
| Training | | | | |
| Mix-Precision | TF32&BF16 | TF32&BF16 | TF32&BF16 | TF32&BF16 |
| Epoch | 1 | 1 | 1 | 1 |
| Batch Size | 128 | 128 | 128 | 128 |
| Max Length | 512 | 512 | 512 | 512 |
| Weight Decay | 0 | 0 | 0 | 0 |
| Warm-up Ratio | 0.03 | 0.03 | 0.03 | 0.03 |
| Learning Rate | 2e-5 | 2e-5 | 2e-5 | 2e-5 |
| Grad Clip | 1.0 | 1.0 | 1.0 | 1.0 |
| $\beta_1$ | 0.85 | 0.85 | 0.85 | 0.85 |
| $\beta_2$ | 0.85 | 0.85 | 0.85 | 0.85 |
| $r_{init}$ | 128 | 64 | 64 | 64 |
| $r_{target}$ | 128 | 64 | 64 | 64 |
| $t_{init}$ | 100 | 100 | 100 | 100 |
| $t_{final}$ | 600 | 600 | 600 | 600 |
| Evaluating | | | | |
| Precision | BF16 | BF16 | BF16 | FP32 |
| Sampling | False | | | |
| Top-P | 1.0 | | | |
| Max Length | GSM-8K: 1024 MATH: 2048 HumanEval: 2048 | | | |

Table 4: Hyperparameters of our experiments of AdaLoRA on different models for GSM-8K and MATH

```
1 Below is an instruction that describes a task. Write a response that
     appropriately completes the request.
2
3 ### Instruction:
4 {question}
5
6 ### Response: Let's think step by step.
```

Listing 1: Prompt used for GSM-8K and MATH.

```
1 @@ Instruction
2 Here is the given code to do completion:
3 ```python
4 {question}
5 ```
6
7 Please continue to complete the function with python programming
     language. You are not allowed to modify the given code and do the
     completion only.
8
9 Please return all completed codes in one code block.
10 This code block should be in the following format:
11 '''python
12 # Your codes here
13 '''
14
15 @@ Response
```

Listing 2: Prompt used for HumanEval evaluation.

## C  PROOFS

**Theorem 5.1.** *The regularizer $\mathcal{L}_{reg}$ is convex.*

*Proof.* We prove this in two steps:

First, we show that $f(U_p) = \|U_p^\top U_p - I\|_F$ is convex. Then, we prove that $g(V_p) = \|V_p^\top V_p - I\|_F$ is convex.

Since the sum of convex functions is convex, this will establish the convexity of $\mathcal{L}_{reg}$.

Let $U_p, W \in \mathbb{R}^{m \times r}$. The Hessian of $f$ at $U_p$ in the direction $W$ is given by

$$
\begin{aligned}
\nabla^2 f(U_p)[W, W] &= \lim_{\epsilon \to 0} \frac{1}{\epsilon^2} \Big( f(U_p + \epsilon W) - 2f(U_p) + f(U_p - \epsilon W) \Big) \\
&= \lim_{\epsilon \to 0} \frac{1}{\epsilon^2} \Big( \|(U_p + \epsilon W)^\top (U_p + \epsilon W) - I\|_F \\
&\quad - 2\|U_p^\top U_p - I\|_F + \|(U_p - \epsilon W)^\top (U_p - \epsilon W) - I\|_F \Big) \\
&= \lim_{\epsilon \to 0} \frac{1}{\epsilon^2} \Big( \|U_p^\top U_p + \epsilon(U_p^\top W + W^\top U_p) + \epsilon^2 W^\top W - I\|_F \\
&\quad - 2\|U_p^\top U_p - I\|_F + \|U_p^\top U_p - \epsilon(U_p^\top W + W^\top U_p) + \epsilon^2 W^\top W - I\|_F \Big) \\
&= 2\|W^\top W\|_F.
\end{aligned}
\tag{16}
$$

Since $\|W^\top W\|_F \geq 0$ for all $W$, we have $\nabla^2 f(U_p)[W, W] \geq 0$, which proves that $f$ is convex.

The proof for $g(V_p)$ follows the same steps as for $f(U_p)$, leading to the same conclusion.

Therefore, both $f(U_p)$ and $g(V_p)$ are convex, and consequently, $\mathcal{L}_{reg}$ is convex. $\square$

**Theorem 5.2.** *The gradient of the regularizer $\mathcal{L}_{reg}$ is Lipschitz continuous.*

*Proof.* Because $U_p$ and $V_p$ are decomposed from $\mathbb{R}^{m,n}$, we could assume that the Frobenius norms of $U_p$ and $V_p$ are bounded, i.e., $\|U_p\|_F \le M_U$ and $\|V_p\|_F \le M_V$, where $M_U$ and $M_V$ are positive constants.

To prove Lipschitz continuity, we need to show that there exists a constant $L > 0$ such that for any two pairs of matrices $(U_{p,1}, V_{p,1})$ and $(U_{p,2}, V_{p,2})$:

$$|\mathcal{L}_{reg}(U_{p,1}, V_{p,1}) - \mathcal{L}_{reg}(U_{p,2}, V_{p,2})| \le L(\|U_{p,1} - U_{p,2}\|_F + \|V_{p,1} - V_{p,2}\|_F) \qquad (17)$$

First, consider $|\|U_{p,1}^\top U_{p,1} - I\|_F - \|U_{p,2}^\top U_{p,2} - I\|_F|$:

$$
\begin{aligned}
|\|U_{p,1}^\top U_{p,1} - I\|_F - \|U_{p,2}^\top U_{p,2} - I\|_F| &\le \|(U_{p,1}^\top U_{p,1} - I) - (U_{p,2}^\top U_{p,2} - I)\|_F \\
&= \|U_{p,1}^\top U_{p,1} - U_{p,2}^\top U_{p,2}\|_F \\
&= \|U_{p,1}^\top U_{p,1} - U_{p,1}^\top U_{p,2} + U_{p,1}^\top U_{p,2} - U_{p,2}^\top U_{p,2}\|_F \\
&\le \|U_{p,1}^\top (U_{p,1} - U_{p,2})\|_F + \|(U_{p,1}^\top - U_{p,2}^\top) U_{p,2}\|_F \\
&\le \|U_{p,1}^\top\|_F \|U_{p,1} - U_{p,2}\|_F + \|U_{p,1} - U_{p,2}\|_F \|U_{p,2}\|_F \\
&\le (\|U_{p,1}\|_F + \|U_{p,2}\|_F)\|U_{p,1} - U_{p,2}\|_F
\end{aligned}
$$
$$(18)$$

Here, we've used the triangle inequality and the sub-multiplicative property of the Frobenius norm. Similarly for $V$:

$$|\|V_{p,1}^\top V_{p,1} - I\|_F - \|V_{p,2}^\top V_{p,2} - I\|_F| \le (\|V_{p,1}\|_F + \|V_{p,2}\|_F)\|V_{p,1} - V_{p,2}\|_F \qquad (19)$$

Combining these results:

$$
\begin{aligned}
|\mathcal{L}_{reg}(U_{p,1}, V_{p,1}) - \mathcal{L}_{reg}(U_{p,2}, V_{p,2})| &\le (\|U_{p,1}\|_F + \|U_{p,2}\|_F)\|U_{p,1} - U_{p,2}\|_F \\
&\quad + (\|V_{p,1}\|_F + \|V_{p,2}\|_F)\|V_{p,1} - V_{p,2}\|_F \\
&\le \max(\|U_{p,1}\|_F + \|U_{p,2}\|_F, \|V_{p,1}\|_F + \|V_{p,2}\|_F) \\
&\quad \cdot (\|U_{p,1} - U_{p,2}\|_F + \|V_{p,1} - V_{p,2}\|_F)
\end{aligned}
$$
$$(20)$$

Let $L_{reg} = \max(\|U_{p,1}\|_F + \|U_{p,2}\|_F, \|V_{p,1}\|_F + \|V_{p,2}\|_F)$. This $L$ is finite because $\|U_p\|_F \le M_U$ and $\|V_p\|_F \le M_V$.

Therefore, we have shown that:

$$|\mathcal{L}_{reg}(U_{p,1}, V_{p,1}) - \mathcal{L}_{reg}(U_{p,2}, V_{p,2})| \le L_{reg}(\|U_{p,1} - U_{p,2}\|_F + \|V_{p,1} - V_{p,2}\|_F). \qquad (21)$$

This proves that $\mathcal{L}_{reg}$ is Lipschitz continuous with Lipschitz constant $L_{reg}$. $\qquad\square$

**Theorem 5.3.** *For convergence of gradient descent, the learning rate $\eta_d$ and regularization parameter $\gamma$ should*

$$\gamma \propto \frac{1}{\eta_d}. \qquad (11)$$

*Proof.* Recall Equation (6), the updating method of SORSA adapters

$$W_{p,t+1} = W_{p,t} - \eta_t \left( \nabla_{W_{p,t}} \mathcal{L}_{train} + \frac{\gamma}{\eta_d} \nabla_{W_{p,t}} \mathcal{L}_{reg} \right).$$

the gradient descent convergence condition will become

$$\eta_t < \frac{2}{L}, \tag{22}$$

where $L$ is a Lipschitz constant. For a SORSA adapter to converge, we need

$$\eta_t < \frac{2}{L} = \frac{2}{L_{train} + \frac{\gamma}{\eta_d} L_{reg}}. \tag{23}$$

Since $\eta_t$ is bounded by $\eta_t \le \eta_d$, for a SORSA adapter to converge during the entire training process, we need to bound the inequality by

$$\eta_t \le \eta_d < \frac{2}{L_{train} + \frac{\gamma}{\eta_d} L_{reg}}. \tag{24}$$

Rearranging this inequality, we get

$$\begin{aligned}
\eta_d(L_{train} + \frac{\gamma}{\eta_d} L_{reg}) &< 2 \\
\eta_d L_{train} + \gamma L_{reg} &< 2 \\
\gamma L_{reg} &< 2 - \eta_d L_{train} \\
\gamma &< \frac{2 - \eta_d L_{train}}{L_{reg}}.
\end{aligned} \tag{25}$$

We can assume that the regularizer's gradients scale with $\eta_d$, meaning that a larger updating step (due to a larger $\eta_d$) will lead to more significant deviations from orthonormality, which increases $L_{reg}$. Conversely, smaller steps lead to a more gradual progression towards orthonormality, which reduces $L_{reg}$. Therefore, we could assume $L_{reg} \propto \eta_d$. Moreover, the $\gamma$ must not be negative, or the regularization term would negatively impact its supposed purposes. Therefore, we can rewrite the inequality as

$$0 \le \gamma < \frac{2}{k\eta_d} - L_{train}, \tag{26}$$

where $k$ is a constant.

Therefore,

$$\gamma \propto \frac{1}{\eta_d}. \tag{27}$$

$\square$

**Lemma 5.4.** *Let $W_p^{unreg} = U_p^{unreg} diag(S_p)^{unreg} (V_p^{unreg})^\top$ be the $W_p$ only training without using regularizer, and $W_p^{reg} = U_p^{reg} diag(S_p)^{reg} (V_p^{reg})^\top$ be the $W_p$ training with the regularizer. For each singular value $\sigma_i$, the following bound holds:*

$$(1 - \epsilon)\sigma_i^{unreg} \le \sigma_i^{reg} \le (1 + \epsilon)\sigma_i^{unreg}, \tag{12}$$

*where $\epsilon$ is a small positive constant, $\sigma_i^{reg}$ and $\sigma_i^{unreg}$ are singular values in the case of training with and without regularizer, respectively.*

*Proof.* First, let's consider the effect of the orthonormal regularizer. The regularizer aims to make $U_p^\top U_p \approx I$ and $V_p V_p^\top \approx I$. We can quantify this approximation as:

$$\|\nabla_{W_p} \mathcal{L}_{reg}\|_F \le \epsilon_\nabla. \tag{28}$$

where $\epsilon_\nabla > 0$ is a small constant.

Then, we define two cases of one-step optimized $W_p$, that $W_p^{\text{reg}}$ is optimized with regularizer, and $W_p^{\text{unreg}}$ is optimized without regularizer.

From Equation (5)

$$W_{p,t+1} = W_{p,t} - \eta_t \nabla_{W_{p,t}} \mathcal{L}_{train} - \gamma_t \nabla_{W_{p,t}} \mathcal{L}_{reg},$$

we could get

$$W_p^{\text{reg}} - W_p^{\text{unreg}} = \gamma \nabla_{W_p} \mathcal{L}_{reg}. \tag{29}$$

Calculating Frobenius norm on both sides, we could find

$$\|W_p^{\text{reg}} - W_p^{\text{unreg}}\|_F = \gamma \|\nabla_{W_p} \mathcal{L}_{reg}\|_F = \gamma \epsilon_\nabla. \tag{30}$$

Now, we can use Weyl's inequality (Weyl, 1912), which states that for matrices A and B:

$$|\sigma_i(A + B) - \sigma_i(A)| \leq \|B\|_2 \leq \|B\|_F. \tag{31}$$

Applying this to our case, with $A = W_p^{\text{unreg}}$ and $B = W_p^{\text{reg}} - W_p^{\text{unreg}}$

$$|\sigma_i^{\text{reg}} - \sigma_i^{\text{unreg}}| \leq \|W_p^{\text{reg}} - W_p^{\text{unreg}}\|_F \leq \gamma \epsilon_\nabla. \tag{32}$$

Let $\epsilon = \gamma \epsilon_\nabla$. Then we have

$$-\epsilon \leq \sigma_i^{\text{reg}} - \sigma_i^{\text{unreg}} \leq \epsilon, \tag{33}$$

rearranging this inequality gives us our desired bound

$$(1 - \epsilon)\sigma_i^{\text{unreg}} \leq \sigma_i^{\text{reg}} \leq (1 + \epsilon)\sigma_i^{\text{unreg}}. \tag{34}$$

$\square$

**Theorem 5.5.** *The orthonormal regularizer in SORSA can improve the condition number of the optimization problem throughout training under certain conditions. Specifically, at initialization* $(t = 0)$*:*

$$\kappa(W_{p,0}^{reg}) = \kappa(W_{p,0}^{unreg}), \tag{13}$$

*where* $\kappa(W_p)$ *denotes the condition number of* $W_p$*;* $W_{p,t}^{reg}$ *and* $W_{p,t}^{unreg}$ *represent* $W_p$ *at time-step* $t$ *in the case of training with or without regularizer, respectively.*

$\exists c > 0$*, while* $t > c$*,*

$$\kappa(W_{p,t}^{reg}) < \kappa(W_{p,t}^{unreg}). \tag{14}$$

*Proof.* Let $W_{p,t} = U_{p,t} \text{diag}(S_{p,t}) V_{p,t}^\top$ be the principal part of the singular value decomposition approximation of $W$ at time-step $t$. The condition number is given by

$$\kappa(W_{p,t}) = \frac{\sigma_{\max}(W_{p,t})}{\sigma_{\min}(W_{p,t})}, \tag{35}$$

where $\sigma_{\max}$ and $\sigma_{\min}$ are the maximum and minimum singular values of $W_{p,t}$.

At initialization $(t = 0)$: Due to SVD initialization, $U_{p,0}$ and $V_{p,0}^\top$ are perfectly orthonormal, so

$$\kappa(U_{p,0}^{\text{unreg}}) = \kappa((V_{p,0}^{\text{unreg}})^\top) = \kappa(U_{p,0}^{\text{reg}}) = \kappa((V_{p,0}^{\text{reg}})^\top) = 1, \tag{36}$$

and $\epsilon_0 = 0$, $\delta_{1,0} = \delta_{2,0} = 0$. Therefore

$$\frac{\kappa(W_{p,0}^{\text{reg}})}{\kappa(W_{p,0}^{\text{unreg}})} = \frac{\kappa(\text{diag}(S_{p,0})^{\text{reg}})}{\kappa(\text{diag}(S_{p,0})^{\text{unreg}})} = 1. \tag{37}$$

During training $(t > 0)$: As training progresses, $U_{p,t}$ and $V_{p,t}^\top$ deviate from orthonormality in the unregularized case. We quantify this deviation:

$$\|U_{p,t}^\top U_{p,t} - I\|_F \leq \epsilon_{1,t} \tag{38}$$

$$\|V_{p,t}^\top V_{p,t} - I\|_F \leq \epsilon_{2,t}, \tag{39}$$

where $\epsilon_{1,t}, \epsilon_{2,t} > 0$ are two constants increase over time t.

For the regularized matrices, we can bound their condition numbers:

$$\kappa(U_{p,t}^{\text{reg}}) \leq 1 + \delta_{1,t}; \tag{40}$$

$$\kappa(V_{p,t}^{\text{reg}}) \leq 1 + \delta_{2,t}, \tag{41}$$

where $\delta_{1,t}, \delta_{2,t}$ are small positive numbers that remain bounded due to the regularization.

From the Lemma 5.4, we arrive at:

$$\frac{(1 + \delta_{1,t})(1 + \delta_{2,t})(\frac{1-\epsilon_t}{1+\epsilon_t})}{\kappa(U_p^{\text{unreg}}) \cdot \kappa((V_{p,t}^{\text{unreg}})^\top)} \leq \frac{\kappa(W_{p,t}^{\text{reg}})}{\kappa(W_{p,t}^{\text{unreg}})} \leq \frac{(1 + \delta_{1,t})(1 + \delta_{2,t})(\frac{1+\epsilon_t}{1-\epsilon_t})}{\kappa(U_{p,t}^{\text{unreg}}) \cdot \kappa((V_{p,t}^{\text{unreg}})^\top)}. \tag{42}$$

As training continues, in the unregularized case, $\kappa(U_{p,t}^{\text{unreg}})$ and $\kappa((V_{p,t}^{\text{unreg}})^\top)$ tend to increase as $U_p$ and $V_p^\top$ deviate further from orthonormality. On the other hand, $(1 + \delta_{1,t})(1 + \delta_{2,t})(\frac{1+\epsilon_t}{1-\epsilon_t})$ will approach to 1 because of the reinforcement in orthonormality will leads to a smaller $\delta_{1,t}, \delta_{2,t}$ and $\epsilon_t$. Therefore, $\exists c > 0$, while $t > c$,

$$\kappa(U_{p,t}^{\text{unreg}}) \cdot \kappa((V_{p,t}^{\text{unreg}})^\top) > (1 + \delta_{1,t})(1 + \delta_{2,t})(\frac{1 + \epsilon_t}{1 - \epsilon_t}), \tag{43}$$

will hold.

Therefore, while $t > c$, we have

$$\frac{\kappa(W_{p,t}^{\text{reg}})}{\kappa(W_{p,t}^{\text{unreg}})} < 1, \tag{44}$$

that indicates an improvement in the condition number. $\qquad \square$

