# OpenReview forum: "SORSA: Singular Values and Orthonormal Regularized Singular Vectors Adaptation of Large Language Models"
_ICLR.cc/2025/Conference — ICLR 2025 Conference Withdrawn Submission_

### Official Review · Reviewer_i8oY · 2024-10-16

**Soundness:** 2
**Presentation:** 2
**Contribution:** 2
**Rating:** 3
**Confidence:** 3

**Summary:**

The authors propose Singular Values and Orthonormal Regularized Singular Vectors Adaptation (SORSA) for parameter-efficient fine-tuning (PEFT). They use an analysis of singular values and vectors to show the limitation of existing works and propose to solve it via orthonormal regularization. Theoretical analysis shows that the condition number of the low-rank incremental matrix is improved via regularization, resulting in enhanced stability. The superior performance of SORSA is validated on three NLG tasks compared to other PEFT methods.

**Strengths:**

1. While orthonormal regularization on incremental low-rank matrices has been used in existing works, the analysis of the condition number is novel to me, and the improved stability makes sense.
2. The approach is simple and effective, as it directly applies orthonormal regularization.

**Weaknesses:**

1. The method is not well-motivated. The authors begin with an analysis of singular values and vectors, which shows a different updating pattern of SORSA compared to other methods. However, it is unclear how this is connected to the limitation of the generalization ability of LoRA and FT. I can only observe a limitation of the learning capacity of LoRA and FT. Additionally, it is not clear why the orthonormal regularization leads to a different updating pattern, as shown in Figure 2, and why this pattern can give an improvement. I suggest a further theoretical justification for this point.
2. There seems to be a misuse of terminology: the authors use FT to denote "partial fine-tuning" (for example, page 2, line 86). However, in the literature, FT often denotes "full fine-tuning" (for example, in DoRA [1]). If the analysis in Sec. 3.2 is truly inspired by DoRA, the authors may want to compare the updating patterns of "full fine-tuning, LoRA, and SORSA" instead of "partial fine-tuning, LoRA, and SORSA." If I misunderstood, could you provide a definition of partial fine-tuning early in the main paper?
3. The experimental comparison is not extensive enough in terms of benchmarks. Only NLG tasks are used to evaluate the method, and it's not clear whether SORSA works in other NLP tasks. More experiments on other tasks, such as the common GLUE benchmark [2] in natural language understanding (NLU), are expected.
4. It seems SORSA (w/o reg) is essentially the same as PiSSA as described in Related Work, so I encourage the authors to use consistent terminology throughout the paper. If they are different, please give a clear explanation of the differences. Also, in this case, quantitative ablation studies are missing since the only comparison between SORSA and SORSA (w/o reg) is in Figure 2. The performance of SORSA (w/o reg) should also be provided in Table 1.



[1] Shih-yang Liu, Chien-Yi Wang, Hongxu Yin, Pavlo Molchanov, Yu-Chiang Frank Wang, Kwang-
Ting Cheng, and Min-Hung Chen. DoRA: Weight-Decomposed Low-Rank Adaptation.
In Forty-first International Conference on Machine Learning, June 2024.

[2] Alex Wang, Amanpreet Singh, Julian Michael, Felix Hill, Omer Levy, and Samuel R Bowman.
Glue: A multi-task benchmark and analysis platform for natural language understanding. arXiv
preprint arXiv:1804.07461, 2018.

**Minor comments**

1. Figure quality can be improved. For example, the font size should be larger for better readability.

**Questions:**

1. Is there any insight into the "Grad Norm" figures in Figure 3?

---

> ### Author Response · Authors · 2024-11-17
>
> Thanks for your review and for acknowledging SORSA's strengths! I also appreciate your feedback on clarifying our motivation and experimental setup, which I will address thoroughly.
>
> 1. The "restriction" I mentioned here isn't the actual optimization restriction. Instead, it's the "restriction" throughout all layers. Figure 2 shows that almost all FT, LoRA, and SORSA layers without regularizer present a synchronized and linear-like updating in $\Delta D$ and $\Delta \Sigma$. This showed that all layers are "locked" with each other, which I explained as "restriction." Although SORSA uses one more regularizer during training, the result in Figure 2 actually presented its more "freely" updating (which is evident that different layers can update much more independently) and reduced updating in $\Delta D$ and $\Delta \Sigma$. Since SORSA eventually achieved an even lower loss, I concluded that FT, LoRA, and SORSA without regularizer actually did more "unnecessary" updates without actually helping the convergence, and I believe it is reasonable to say that those "unnecessary" updates will have more disruptive effects on pre-trained well-generalized LLM than SORSA, which have much less unnecessary updating.
> 2. I apologize for the confusion here. (FT) here was a typo. Here should be (Partial FT). Partial FT only trains some matrices instead of full FT training for the whole model. I was comparing partial fine-tuning, LoRA, and SORSA because they have similar trainable parts (Although LoRA and SORSA are training the Low-rank part of it).
> 3. I agree that the experimental scope can be broadened. Initially, I conducted NLU tests using RoBERTa and DeBERTa on the GLUE benchmark. However, I found that the results for all methods (LoRA, PiSSA, FT, and SORSA) varied significantly depending on the random seeds and hyperparameters. Different seeds consistently produced completely inconsistent outcomes. Currently, other papers that still focus on NLUs typically rely on grid searches to optimize hyperparameters, making these experiments heavily dependent on time and luck rather than genuinely showcasing the potential of the methods. Moreover, NLU models and tasks have recently lost some of their popularity. As a result, instead of presenting confusing results or spending days on grid searches to achieve a high metric in a random epoch just to highlight my method, I decided to skip the NLU part.
> 4. SORSA (w/o reg) is similar to PiSSA. However, PiSSA merged $\text{diag}(S)$ into $U$ and $V^\top$, where SORSA (w/o reg) keep them separatly. However, during optimization, they work essentially the same way. The reason I stressed SORSA (w/o reg) in Figure 2 is that I want to stress the importance of regularizers instead of letting readers guess why they work differently. In Table 1, I want readers to focus more on the comparison of different methods instead of one method with different setups.
> For the grad norm, I added the comparison of the changing rate of curves.
>
> I will also work on fixing figures I can barely read without zooming in.
>
> Thanks again for providing helpful reviews for my paper.

---

> ### Author Response · Authors · 2024-12-02
>
> Deer Reviewer,
>
> I submitted my rebuttal some time ago and uploaded an updated PDF document. I noticed that there hasn't been any feedback yet. I would greatly appreciate if you could take a moment to review my responses to your initial comments and the updated PDF, as they address several important points raised in the reviews. Thank you for your time and consideration.
>
> Best regards,
> Author of Submission805

---

### Official Review · Reviewer_R9g8 · 2024-11-03

**Soundness:** 2
**Presentation:** 2
**Contribution:** 3
**Rating:** 6
**Confidence:** 4

**Summary:**

The SORSA method proposed in this paper is a variant of LoRA—a technique for parameter-efficient fine-tuning of pre-trained large language models (LLMs). SORSA combines concepts from two previously proposed LoRA variants: AdaLoRA (Zhang et al., 2023) and PiSSA (Meng et al., 2024). In the original LoRA method, each weight matrix $W$ in an LLM is adapted as $W+AB$, where $AB$ represents the product of two low-rank matrices, $A$ and $B$, which are the trainable parameters. AdaLoRA adapts the weight matrices as $W+USV^T$, where $U$ and $V$ are randomly initialized low-rank matrices, and $S$ is a zero-initialized diagonal matrix. During training, $U$ and $V$ are regularized to form orthonormal bases in their columns, so that $USV^T$ resembles the singular value decomposition (SVD) of a matrix. Similar regularization has been shown to speed up training in many previous works.

PiSSA, on the other hand, directly decomposes each pre-trained weight matrix using SVD as $W=USV^T$. During fine-tuning, only the parameters associated with the largest singular values and the corresponding singular vectors (selected columns of $U$ and $V$) are updated, while the remaining parameters remain fixed. PiSSA does not use any orthogonality-enforcing regularization during fine-tuning. SORSA, the method proposed in this paper, essentially applies PiSSA but regularizes the fine-tuned vectors (selected columns of $U$ and $V$) to stay nearly orthonormal, as in AdaLoRA.

**Strengths:**

The paper reports superior performance for LLMs fine-tuned with SORSA compared to the PiSSA method. Additionally, the proposed approach is relatively easy to implement. The originality of this work lies in combining the ideas behind AdaLoRA and PiSSA, as described in the "Summary" section.

**Weaknesses:**

Firstly, the paper suffers from poor writing quality, with numerous grammatical errors and awkward English expressions (too many to list exhaustively in this review). The text is also poorly structured, with some mathematical symbols left undefined. The figures use such small font sizes that they are almost unreadable, even in the electronic version where one can zoom in. More detailed feedback is provided below. The paper appears to be written by an inexperienced author, so I would recommend seeking assistance from someone with more experience and strong English skills to review and refine the text.

Besides the standard full fine-tuning and LoRA, the paper compares the proposed SORSA method only with PiSSA. However, since SORSA builds on AdaLoRA’s concepts, I believe AdaLoRA (and possibly other related methods like OLoRA or DoRA) should also be included in the comparisons. Furthermore, the experimental results are limited compared to other referenced works (e.g., the PiSSA paper).

The author’s limited experience is further evident in presenting some trivial findings as if they were major contributions. For example, Section 4 refers to Appendix A, where readers are promised an optimized and highly efficient version of SORSA. However, Appendix A merely states the obvious: that multiplying by a diagonal matrix is more efficient than naively multiplying by a full matrix with zeroed-out off-diagonal elements. The appendix also includes an analysis of the speedup achieved by this "optimized" implementation, but it yields nonsensical results that do not align with the $O(N^2)$ versus $O(N^3)$ complexity of the optimized and naive approaches. Similarly, the statement that the regularizer is convex and Lipschitz continuous is rather obvious, although the corresponding derivations are reasonably included in an appendix.

**Questions:**

The paper should clarify its relationship to AdaLoRA and PiSSA in the "Related Works" section (or even the Introduction), which it currently does not.

There appear to be inconsistencies in notation with symbols like $V_{[:r]}$, $U_{[:r,:r]}$ and $V_{[:r,:]}$ in Section 2. Additionally, "range" notation is introduced only in Section 4.

Section 3.1 explains SVD, but this explanation is not well-written and is confusing. Given that SVD is a standard technique, it is unnecessary to explain it to the paper's target audience. Simply introducing the notation should suffice.

What is $\Sigma$ in Section 3.1? It is not defined. Is it the diagonal matrix of singular values? The paper already uses $S$ for the vector of singular values.

It may be clearer to use $\mathrm{diag}(S)$ instead of $\mathrm{diag}(W)$ at the end of page 3 and the symbol $W$ is already dedicated to weight matrices.

Section 3.2, titled "Analysis Method," does not actually describe any analysis method. Instead, it introduces metrics that express the deviation of the singular values and vectors of the updated weight matrices from those of the original pre-trained matrices. However, metrics alone do not constitute an analysis method. Moreover, these ad-hoc metrics and the conclusions drawn from their behavior in Figure 2 are speculative, with no analysis showing a direct correlation between these metrics and fine-tuning performance.

"$\Delta\Sigma_t$ represents singular value variants between $W_0$ and $W_t$"
should perhaps read
"$\Delta\Sigma_t$ represents the distance/difference between singular values of $W_0$ and $W_t$"

Section 3.3 compares the behavior of metrics from Section 3.2 for various fine-tuning methods, including SORSA. However, since SORSA has not yet been introduced, readers may not understand why certain behavior is expected. Therefore, this analysis should appear later in the paper, possibly in the appendix, as it is speculative and not essential for understanding the main advantages of the proposed technique.

"significant adjustments in significant vectors" should read "significant adjustments in singular vectors"

"parallel updating pattern across weights in different layers, which emphasizes a restriction of these methods"
I would say that it shows that there is not any restriction on updating parameters in all layers.

"indicating that the updates in the SORSA are less constrained"
Since the changes in singular values and vectors are smaller for SORSA, it suggests that the updates are actually more constrained, as SORSA uses orthonormality regularization as an additional constraint.

"matrix that preserves the largest significant values and vectors, containing the matrix’s most significant data"
Again, "significant" should likely be "singular". What does it mean "the matrix’s most significant data"? This is very poor description of what the largest singular values and the corresponding singular vectors represent.

"which consist $W_p$" should perhaps read "which constitute $W_p$" ... and similarly for $W_r$.

Equation (10) presents an implementation detail that seems unnecessary for inclusion in the main paper.

What is $k$ in Equation (11)? Without defining it, Equation (11) implies that $\gamma$ is smaller than any arbitrary number.

"... SORSA, we present a novel analysis of its condition number"
In (7), SORSA is presented as a function, so it does not have a condition number. The matrix $W_r+W_p$ does.

What are $\sigma_i^{unreg}$ and $\sigma_i^{reg}$ in Equation (12)? Are they the singular values of the fine-tuned models? The derivation of Equation (12) in the appendix is difficult to follow. Where does Equation (25) originate? What do $W_p^{unreg}$  and $W_p^{reg}$ represent in this equation—the parameters of the regularized and unregularized fine-tuned model? Equation (25) implies that the Frobenius norm of the differences between in trained matrices $W_p^{unreg}$  and $W_p^{reg}$ equals the scaled Frobenius norm of the gradient of the regularizer with respect to a $W_p$ matrix. Which $W_p$ matrix? What point is the gradient evaluated at? Equation (25) is completely unclear.

The symbols  $\delta_{1,t}$ and $\delta_{2,t}$ are not introduced in Theorem 5. Likewise, symbols $\kappa$, $\sigma_{max}$ and $\sigma_{min}$ are not introduced, though their meanings may be guessed.



**===End of the original review ====**

Here, I provide response to the Authors comments on the review as I was not able to respond by the official means:


The paper has been significantly improved in the current updated version. It is much easier to follow thanks to its improved structure and presentation. Additionally, results for AdaLoRA have been added, making the results more convincing. Accordingly, I am increasing my scores for the paper.

However, several issues remain. Some of these were already pointed out in the reviews but have not been addressed in the updated paper:

Multiple reviewers requested clarification on the relationship between SORSA, AdaLoRA, and PiSSA. Specifically, we asked for acknowledgment that SORSA is primarily a combination of ideas from these two methods. This relationship (especially with AdaLoRA) is still not clearly stated in the paper.

One reviewer pointed out that "SORSA (w/o reg) is essentially the same as PiSSA." I agree with this assessment, and it should be explicitly stated in the paper. While I understand that PiSSA merges singular values with the matrices of singular vectors, this correspond to only different parametrization of the same model and should not significantly affect optimization (as you confirm in your response). You mentioned that you avoided explicitly stating this to prevent confusion, but I believe that not clearly acknowledging the equivalence of "SORSA (w/o reg)" and "PiSSA" is actually more misleading.

I still find the "Singular Values and Vector Analysis" section speculative and of limited scientific value. As I noted in my previous review, "the ad-hoc metrics and the conclusions drawn from their behavior in Figure 2 are speculative, with no analysis showing a direct correlation between these metrics and fine-tuning performance." Why must we examine differences between weight matrices in terms of singular values and singular vectors separately? What do the patterns in Figure 2 teach us? Low-rank approximation using SVD minimizes the Frobenius norm of the difference between the original matrix and the approximating matrix, so why not simply measure similarity using the Frobenius norm? Wouldn't this metric be equally expressive? For example, the last graph in Figure 2 shows that in some layers, only singular values change while singular vectors remain constant. Does this observation have any meaningful implications? There is no analysis provided to support its relevance.

You state:

"Figure 2 shows that almost all FT, LoRA, and SORSA layers without a regularizer exhibit synchronized and linear-like updating in singular values and vectors. This shows that all layers are 'locked' with each other, which I interpret as 'restriction.' Although SORSA uses one additional regularizer during training, Figure 2 actually demonstrates its more 'free' updating (evident in how different layers can update more independently)."

What does it mean that "layers are 'locked' with each other"? Are you suggesting that updates are correlated across layers? There is no measurement of such correlations in the paper. Nor do you demonstrate that parameters are not updated independently in FT and LoRA. Figure 2 merely shows that all parameters across all layers are updated in FT and LoRA, while only some layers are effectively updated in SORSA. Perhaps stronger (L2 or L1) regularization toward the original weight matrix would achieve the same effect in LoRA.

I strongly suggest removing the section on the "optimized version of SORSA," even from the appendix. The optimization you describe is the only reasonable way to implement multiplication with a diagonal matrix, something anyone with basic algebra knowledge and programming skills would do by default. Admitting that you considered converting the diagonal matrix to a dense one and multiplying with it is embarrassing. Moreover, what you call element-wise multiplication in your equation is not truly element-wise, as the vector must first be broadcast. Finally, as I noted in my previous review, Figure 4 does not illustrate the quadratic vs. cubic complexity as claimed in the text.

The practical purpose of Equation (11), defining $\gamma$ remains unclear. I had hoped your derivation in the appendix would clarify this, but it did not. You state:

"$k$ will be a constant. This essentially implies $\gamma$ should be inversely proportional to $n_d$"

Here, $n_d$ is a constant hyperparameter (maximum learning rate), and you define $\gamma =k / n_d$, where $k$ is another hyperparameter. Why should $k$ be tuned instead of directly adjusting $\gamma$?

It is still unclear what the singular values in Equation (12) represent. Are they the corresponding values (regularized and unregularized) from the respective updates? They do not depend on any iteration index $t$ in the formula.

Minor Issues:

The figures, especially Figure 2, use fonts that are too small to read, although the authors promise to improve them.
Use diag(S) instead of diag(W), as W is already used to denote a weight matrix, not a vector.

The subsection titled "Analysis Method" should be renamed to something like "Measuring Similarity Between Singular Values and Vectors," as the current title does not accurately reflect its content.

---

> ### Author Response · Authors · 2024-11-13
>
> Thank you for your cautious and elaborate review. I am sorry for all the obstacles caused by my poor writing skills in your review process. Your assumptions about weaknesses are absolutely true, and I've acknowledged the issue of writing and phrasing and am actively fixing it.
>
> Due to the limited time before the deadline, I didn't have enough time to conduct further experiments for AdaLoRA. However, I will update the results of AdaLoRA as soon as possible. I indeed compared the metrics presented in PiSSA's work directly in some experiments. However, my experiments referenced their setups carefully, and I also had several email conversations about their experiment details so I could ensure I had the exact same setup with them besides the PEFT method.
>
> Also, I really agree that some theorems and Appendix A seem trivial. I found that the paper on AdaLoRA didn't provide proof of convexity and Lipschitz's continuity, which may have been due to its being too trivial. However, I want to present as much detail as possible for a comprehensive work. For Appendix A, I didn't clarify the subject of complexity, which I was only comparing the time complexity of $U_p \odot S_p$ with $U_p S_p$. Thanks so much for pointing out this ambiguity.
>
> I will respond to all the following questions in points:
>
> Line 1. Thanks for pointing that out. I should have included our connections more explicitly.
>
> Line 2. Thanks for sharing that flaw in my writing. I have now briefly included the information on notation in Section 2.
>
> Line 3-5. I removed the brief introduction of SVD and clarified the definition of $diag(\cdot)$.
>
> Line 6, 10, and 11. The "restriction" I mentioned here isn't the actual optimization restriction. Instead, it's the "restriction" throughout all layers. Figure 2 shows that almost all FT, LoRA, and SORSA layers without regularizer present a synchronized and linear-like updating in $\Delta D$ and $\Delta \Sigma$. This showed that all layers are "locked" with each other, which I explained as "restriction." Although SORSA uses one more regularizer during training, the result in Figure 2 actually presented its more "freely" updating (which is evident that different layers can update much more independently) and reduced updating in $\Delta D$ and $\Delta \Sigma$. Since SORSA eventually achieved an even lower loss, I concluded that FT, LoRA, and SORSA without regularizer actually did more "unnecessary" updates without actually helping the convergence, and I believe it is reasonable to say that those "unnecessary" updates will have disruptive effects towards optimization. Although this part is not analyzing mathematical, I think readers could understand SORSA's distinction in optimization pattern, compared with FT, LoRA, and SORSA w/o regularizer. Therefore I call it "analysis".
>
> Line 7. Thanks for pointing out that bug.
>
> Line 8. Thanks for giving this opinion on my so-called analysis. I will work on finding a better way to present this part.
>
> Lines 10, 12, and 13. Thanks for pointing out these poor expressions. I will upload the fixed version in the following updates.
>
> Line 14. This is actually important for the later section of gradient analysis. But maybe I should place it there instead.
>
> Line 15. $k$ will be a constant. This essentially implies $\gamma$ should be inversely proportional to $\eta_d$.
>
> Line 16. Thanks for identifying this issue. It should be "condition numbers of SORSA adapters during the optimization process."
>
> Line 17. They are singular values in the case of optimizing without or with a regularizer, respectively. Eq. 25 is from the definition of $W^{reg}$ and $W^{unreg}$, in which their difference is the learning rate multiply the gradient of the regularizer respected to their parameters. The next version will add how I got it from Eq 9.
>
> Line 18. Thanks for pointing out this ambiguity. These issues will be fixed in the next update.
>
> Again, thanks so much for providing these really insightful and helpful reviews!

---

> ### Author Response · Authors · 2024-12-02
>
> Deer Reviewer,
>
> I submitted my rebuttal some time ago and uploaded an updated PDF document. I noticed that there hasn't been any feedback yet. I would greatly appreciate if you could take a moment to review my responses to your initial comments and the updated PDF, as they address several important points raised in the reviews. Thank you for your time and consideration.
>
> Best regards,
> Author of Submission805

---

### Official Review · Reviewer_j6kg · 2024-11-05

**Soundness:** 2
**Presentation:** 3
**Contribution:** 2
**Rating:** 5
**Confidence:** 4

**Summary:**

This paper introduces SORSA that is a parameter-efficient fine-tuning (PEFT) method designed to adapt large language models (LLMs) for downstream tasks. The experimental results demonstrate  that it can converge faster than PiSSA and LoRA and achieves the higher accuracy on benchmarks like GSM-8K and MATH.

**Strengths:**

1. The proposed method outperforms the other PEFT techniques such as LoRA and PiSSA in terms of accuracy on various benchmarks, showcasing its effectiveness. The results look very promising on a variety of experiments.
2. The authors also analyze the variation patterns of singular values and vectors during parameter updates and compare SORSA with other PEFT methods such as LoRA and partial fine-tuning.

**Weaknesses:**

1. The noverty of this paper is limited. the initialization method is from Pissa [1], and updates in the form of singular value decomposition and the orthonormality regularizer are from AdaLoRA [2].
2. Some symbols in Theorem 3 and Theorem 5 are used without the previous definition, which can be confusing. It is better restate these theorems to make them more straightforward.
3.  Proof of Theorem 2 is not right. Line 1035-1036. "This L is finite because the Frobenius norms
of U and V are bounded (they represent orthonormal matrices in the ideal case)". This statement not ritght. You need to add the condition that Frobenius norms of U and V are bounded. $\mathcal{L}_{reg}$ is Lipshitz only when Frobenius norms of U and V are bounded.
4. Theorem 3 is questionable. Related workes on convergence of optimizers for transformers are not clear. They only give some preliminary results under some strict assumptions and settings. But Therorem 3 needs nothing. The proof of Theorem 3 is based on Eq (21), howerver, it has no basis.
5. Due to my limited time, I haven't checked all the lemmas and theorems, but they do have some problems. I strongly recommend that all authors carefully examine the theoretical analysis to make sure it is correct.
6. In table 1，the result of RWKV6 7B on GSM-8K for LoRa is only 8.04%. This is very strange (too
low). Make sure the experiment setting is correct. Since SORSA is similar to Pissa and AdaLoRA. It is better to campare with the results of AdaLoRA as well.
[1] Fanxu Meng, Zhaohui Wang, and Muhan Zhang. PiSSA: Principal Singular Values and Singular
Vectors Adaptation of Large Language Models.
[2] Qingru Zhang, Minshuo Chen, Alexander Bukharin, Pengcheng He, Yu Cheng, Weizhu Chen, and Tuo Zhao.  Adaptive Budget Allocation for Parameter-Efficient Fine-Tuning. ICLR 2023.

**Questions:**

The same questions as given above in the section of Weaknesses.

---

> ### Author Response · Authors · 2024-11-13
>
> Thank you for your detailed review. I really appreciate your acknowledgment of SORSA’s superior accuracy and faster convergence over other PEFT methods like LoRA and PiSSA. Your feedback on clarity and rigor in the theoretical proofs and comparisons with related methods has been very constructive, and I'm committed to addressing each of your concerns thoroughly.
>
> Q.1: While SORSA builds upon aspects of existing work, my approach represents a unique and meaningful advance. Combining singular value decomposition (SVD) and an orthonormality regularizer in a novel framework achieves accelerated convergence and better generalization on tuned models beyond FT, LoRA, and PiSSA. I also believe the analysis provided a new aspect of analyzing the PEFT training process.
>
> Q.2,3,4,5: Thanks for pointing out these issues! Your comments regarding clarity in Theorems 2, 3, and 5 are very helpful, and I will add restatements for previous equations,  flaws in Theorem 2, and a more apparent derivation of Eq. (21) in the later version. I will also actively work on checking on other lemmas and theorems.
>
> Q.6: I totally agree with your concerns. In fact, in line 436, I addressed this observation and provided my explanation for this result. Due to the limited time before the submission deadline, I haven’t conducted experiments on AdaLoRA yet, but I will update the results as soon as possible.
>
> Finally, I really appreciate all your valuable suggestions and feedback. My updates will enhance the rigor and clarity of my work and address your concerns comprehensively.
>
> Again, thanks for your insightful review.

---

> ### Author Response · Authors · 2024-12-02
>
> Deer Reviewer,
>
> I submitted my rebuttal some time ago and uploaded an updated PDF document. I noticed that there hasn't been any feedback yet. I would greatly appreciate if you could take a moment to review my responses to your initial comments and the updated PDF, as they address several important points raised in the reviews. Thank you for your time and consideration.
>
> Best regards,
> Author of Submission805

---

### Author Response · Authors · 2024-11-18

Deer reviewers,
Thanks for all your elaborate and detailed reviews. Your feedback significantly helped me to polish my work further.

I submitted the rebuttal version of the paper. Which includes updates:
1. The Analysis section is now placed after the SORSA section.
2. I rewrote the analysis results in part and made it more detailed and elaborate.
2. Added more elaborate analysis to the analysis result section.
3. Fixed some undefined variables in theorems and explained what $W^{reg}$ and $W^{unreg}$ is.
4. Removed the SVD geometric definition section and changed it only to introduce my notations.
5. Added upper bound of $U_p$ and $V_p$ in theorem 2.
6. Elaborated the derivation of theorem 3.
7. Added AdaLoRA metrics.
8. Fixed a lot of typos.
9. Added a footnote for one under-performed LoRA metric.
10. Clarify the faster SORSA section.
11. Fixed numerous grammar issues.
12. Let the abstract be more concise.
13. Add explicit declarations of SORSA's connection with PiSSA and AdaLoRA.
14. I added some comparisons to the changing rate of Grad_Norm.
15. Rewrite the Future Work section.

I'm really open to have further discussion on my rebuttal version. If you still have any questions, please do not hesitate to share it.

Again, thanks to you all for giving valuable reviews of my work.

---

### Note · Authors · 2025-01-25

I have read and agree with the venue's withdrawal policy on behalf of myself and my co-authors.